# Variation of Osculating Orbit Elements Using Low-Thrust Photonic Laser Propulsion in the Two-Body Problem

**Fu-Yuen Hsiao**

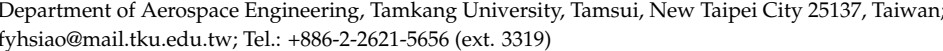

Department of Aerospace Engineering, Tamkang University, Tamsui, New Taipei City 25137, Taiwan; fyhsiao@mail.tku.edu.tw; Tel.: +886-2-2621-5656 (ext. 3319)

**Abstract:** This study investigated the variation of the osculating orbit elements of a spacecraft propelled by photonic laser propulsion (PLP) under the two-body problem assumption. The PLP thrusting system can produce continuous and constant thrust. This paper first reviewed its basics and then studied its influences on the variation of osculating orbit elements given a small PLP thrust. Gauss's equations, perturbation theory, and normalization were introduced to investigate this problem. Our work approached the problem by studying the influences of small planar and out-of-plane PLP thrusts, respectively. Bounds on the variation of orbit elements were derived, and a sufficient condition that traps the mission spacecraft in the vicinity of the mother ship was also found. Numerical simulations are also presented to verify our results, including the bounds and the sufficient conditions. The results obtained in this paper are directly applicable to the usage of PLP thrust, a new type of thrusting system, in the future, and are potentially helpful to various space missions, especially interplanetary travel.

**Keywords:** photonic laser propulsion; Gauss equations; perturbation theory; osculating orbit elements; interplanetary travel





## 1. Introduction

This paper studied the trajectory variation of spacecraft propelled by a low-thrust photonic laser propulsion (PLP) system under the two-body problem environment. In the past few decades, many researchers have shown great interest in the photon thruster and its applications to interplanetary travel, because of its highly efficient and continuous thrusting properties [1–5]. In 2002, Thomas R. Meyer et al. [6] studied the idea of the laser elevator by momentum transfer using an optical resonator. Young K. Bae [7] proposed the concept of photonic laser propulsion in 2008. In that study, the PLP thruster was modeled as a continuous, low-thrust, and high-specific-impulse ($I_{sp}$) engine. A potential application of the PLP-driven missions proposed by these studies is interplanetary travel. A journey to the Alpha Centauri star system was even proposed in 2016 by the Breakthrough Starshot project by Yuri Milner, Stephen Hawking, and Mark Zuckerberg [8].

However, these research works focused more on the PLP thruster itself, and less on the influence on the trajectory. A common assumption is seen in these research works: since the laser continuously pushes and accelerates the mission spacecraft, the mission spacecraft will eventually reach a very high speed, and interplanetary or interstellar travel can be accomplished in a very short traveling time, regardless of the practical problems of the aiming and the relativity effects. It was not until 2011 that Wang and Hsiao [9] investigated the trajectory of the PLP spacecraft under the two-body-problem assumption. Wang and Hsiao realized that some constraints still exist in the applications of PLP thrust [9]. Equilibrium points exist in the planet–mother ship–mission spacecraft system. The mission spacecraft will be trapped in the vicinity of the mother ship if the PLP force is smaller than a threshold, as proposed in [9].

On the other hand, much attention has been focused on a continuous, low-thrust engine, the efficiency of which was proven in the Deep Space 1 mission by NASA [10]

and in a lunar mission by the European Space Agency (ESA) [11]. Several interplanetary missions demonstrated the feasibility of using a low-thrust engine, such as by electric propulsion, as the main propulsion system of the spacecraft. The idea of using the power of light has been studied for decades. Many space missions using light propulsion have been implemented using solar sails. The drawbacks of solar light are that it is passive and its magnitude is inversely proportional to the square of the distance. As a result, the propulsive force decays sharply as long as the spacecraft is far away from the Sun. A continuous and constant PLP force will suppress these drawbacks.

This article is a continuation of the work of [9]. As mentioned earlier, a criterion is imposed on the magnitude of the PLP thrust for practical applications. We were interested more in the influences of a small PLP thrust on the trajectory, because in the current technology, it is infeasible to generate a very large laser power from a space-borne base. By studying the spacecraft trajectory driven by a small PLP thrust, we are able to propose a reasonable laser power criterion for the application of PLP thrust for interplanetary or interstellar travel.

A preliminary work was presented in [12]. The results described here, as an extension of the preliminary work, derived more bounds on the variation of the planar orbit elements, adding a full section discussing the out-of-plane cases and providing many more numerical simulations for its verification. First, this paper briefly reviewed some facts and theories regarding the PLP system. The Gauss equations were employed to understand the variation of the osculating orbit elements. Normalization was also employed to generalize the results. Perturbation theory was applied to study the zeroth- and first-order approximations provided that the mission spacecraft is driven by a small PLP thrust. With certain assumptions, we were able to obtain bounds on the variation of the osculating orbit elements. Numerical simulations are provided to verify the proposed algorithms.

The paper is arranged as follows: Section 2 reviews some facts about PLP thrust; Section 3 reviews the assumptions of the investigated system and the normalization process; Section 4 briefly reviews the Gauss equations and perturbation theory; Section 5 and Section 6 study the variation of the osculating orbit elements by applying planar and out-of-plane thrusting forces, respectively; Section 7 provides some numerical simulations to verity the proposed algorithms; the last section of the paper concludes our work.

## 2. Photonic Laser Propulsion

Many scientists have discussed the concept of using a laser to provide thrust. However, lasers are very inefficient at generating thrust. Although photonic engines have a larger specific impulse compared to conventional ones, they have a smaller thrust-to-power ratio [7]. The specific impulse is approximately $I_{sp} = 3.06 \times 10^7$ s, whereas the thrust-to-power ratio is approximately $T/P = 3.34 \times 10^{-9}$ N/W.

Bae proposed an active resonant optical cavity between two space platforms. In this design, the photon thrust $F$ produced on each mirror is given by [7]:

$$F = \frac{E}{ct},$$ (1)

where $E$ is the energy of each photon, $c$ is the speed of light, and $t$ is the interaction time. By some manipulations, the thrust can be expressed as [7]:

$$F = \frac{2PR_mS}{c},$$ (2)

where $R_m$ is the mirror reflectance and $S$ is the apparent photon-thrust-amplification factor, defined as the ratio of the intracavity laser power to the extracavity laser power $P$. The term $S$ is approximated by:

$$S = \frac{1}{1 - R_m}.$$ (3)

The roof-of-concept demonstration can be found in [7]. Figure 1 presents some experimental data. According to Equations (2) and (3), the thrust is a continuous and stable force. Even though the force is very small, the continuous force keeps driving the spacecraft until it reaches the desired velocity. Figure 2 illustrates the application of PLP to a spacecraft. The launching process starts with a mother ship, which emits a laser beam at the mission ship to generate thrust. Because of the conservation of momentum, however, the mother ship moves in the opposite direction of the mission ship. Thus, a conventional thruster installed on the mission ship must act against the momentum caused by PLP to prevent the mother ship from falling out of orbit.

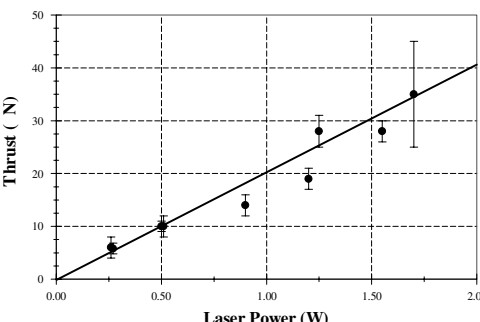

**Figure 1.** Photon thrust data obtained with an output coupler mirror with a reflectance of 0.99967 [7].

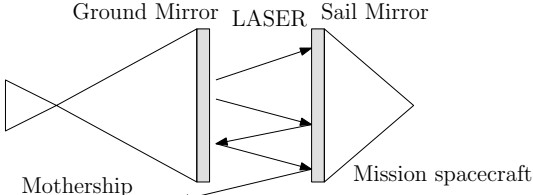

**Figure 2.** Diagram of the photonic laser propulsion system on a spacecraft [6].

## 3. Equations of Motion

Figure 3 shows the relative locations of the central body, mother ship, and mission spacecraft. Let **r** be the position vector of the mission spacecraft, **R** the position vector of the mother ship, and **r** − **R** be the relative position of the spacecraft with respect to the mother ship. Assume the masses of the mother ship and the spacecraft are negligibly small so that they do not produce any gravitational force. According to Newton's gravitation law and Newton's second law of motion, the equations of motion (EOMs) of the mother ship and spacecraft are given by:

$$\ddot{\mathbf{R}} = -\frac{\mu}{R^3}\mathbf{R}, \tag{4}$$

$$\ddot{\mathbf{r}} = -\frac{\mu}{r^3}\mathbf{r} + F\hat{\mathbf{L}}, \tag{5}$$

where $\mu$ is the gravitational parameter of the central body, $\mathbf{L} = \mathbf{r} - \mathbf{R}$, $\hat{\mathbf{L}} = \mathbf{L}/|\mathbf{L}|$, and $F$ is the PLP force given by Equation (2). Note that Equations (4) and (5) are described in the inertial frame (IF).

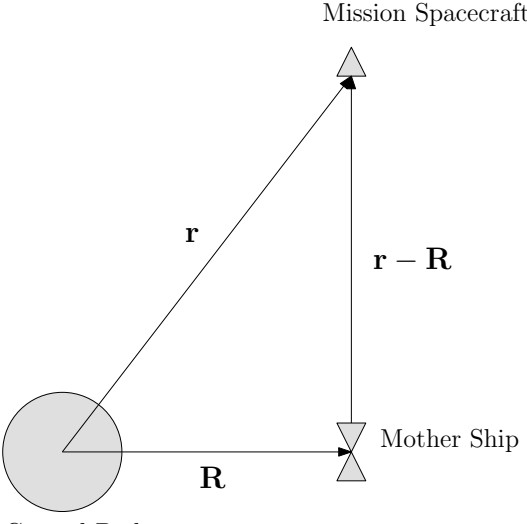

**Figure 3.** Relative positions between the central body, mother ship, and mission spacecraft.

Two coordinate systems were introduced in the inertial frame in the study. First of all, the J2000 Coordinate System, represented by $(\mathbf{I}, \mathbf{J}, \mathbf{K})$, was introduced to specify the orbit elements. In this system, $\mathbf{I}$ points toward the vernal equinox, $\mathbf{K}$ points toward the North Pole of the Earth, and $\mathbf{J}$ completes the triad by the right-hand rule. Although J2000 is defined by the Earth, it is also widely used in the exploration of other celestial bodies. The other coordinate system is represented by $(\mathbf{i}, \mathbf{j}, \mathbf{k})$. In this system, $\mathbf{i}$ points toward the initial position of the mother ship, $\mathbf{k}$ points toward the angular momentum of the mother ship, and $\mathbf{J} = \mathbf{k} \times \mathbf{i}$.

This study did not consider the reaction force by the PLP on the mother ship because in practical applications, we assumed that this reaction is counteracted by a traditional propulsion system. Moreover, this study also assumed that $F$ is constant and acts along the relative position of the mission spacecraft and the mother ship.

## 4. Variation of Osculating Orbit Elements and Perturbation Theory

### 4.1. Osculating Orbit Elements and Normalized Gauss Equations

The motion described by Equation (5) is Keplerian if $F \equiv 0$. A Keplerian trajectory can be described by six canonical orbit elements: $a$ the semi-major axis, $e$ the eccentricity, $i$ the inclination, $\Omega$ the longitude of ascending node, $\omega$ the argument of periapsis, and $M$ the mean anomaly. These elements are functions of the spacecraft position $\mathbf{r}(t)$ and velocity $\mathbf{v}(t)$ in the IF. Without external perturbation forces, the orbit elements remain constant over time.

When an external force is exerted on a spacecraft, the variation of its osculating orbit elements is described by the Gauss-type planetary equations [13]. Moreover, normalization is helpful to generalize applications. In this paper, the only perturbation results from the PLP thrust between the mother ship and the mission ship. Hence, it is natural to normalize this system with the mother ship's parameters.

Assume a spacecraft is suffering from a continuous force $\mathbf{F} = F_r \hat{\mathbf{e}}_r + F_f \hat{\mathbf{e}}_f + F_h \hat{\mathbf{e}}_h$, where $\hat{\mathbf{e}}_r$ and $\hat{\mathbf{e}}_h$ are the unit vectors along the position and angular momentum, respectively, and $\hat{\mathbf{e}}_f = \hat{\mathbf{e}}_h \times \hat{\mathbf{e}}_r$. Define $n_R = \sqrt{\mu/R^3}$ to be the mean motion of the mother ship, and some fundamental parameters are normalized as follows:

$$\tilde{r} = \frac{r}{R}, \tag{6}$$

$$\tau = n_R t, \tag{7}$$

$$\tilde{\mathbf{F}} = \mathbf{F}\frac{R^2}{\mu} = F_r\frac{R^2}{\mu}\hat{\mathbf{e}}_r + F_f\frac{R^2}{\mu}\hat{\mathbf{e}}_f + F_h\frac{R^2}{\mu}\hat{\mathbf{e}}_h$$

$$= \tilde{F}_r\hat{\mathbf{e}}_r + \tilde{F}_f\hat{\mathbf{e}}_f + \tilde{F}_h\hat{\mathbf{e}}_h, \tag{8}$$

$$\tilde{a} = \frac{a}{R}, \tag{9}$$

$$\tilde{n} = \frac{n}{n_R} = \sqrt{\frac{1}{\tilde{a}^3}}. \tag{10}$$

Since $\tilde{n} = \sqrt{\tilde{\mu}/\tilde{a}^3}$, we conclude that $\tilde{\mu} = 1$. The elements $e$, $i$, $\omega$, $\Omega$, and $f$ are already dimensionless. Hence, $\tilde{e} = e$, $\tilde{i} = i$, and so on. Moreover, the derivative of a differential function $X(t)$ with respect to dimensionless time $\tau$ can by found by the chain rule, given by,

$$\frac{dX}{d\tau} = \frac{dX}{dt}\frac{dt}{d\tau} = \frac{1}{n_R}\frac{dX}{dt}. \tag{11}$$

According to these definitions, the normalized parameters and elements of the mother ship are determined as follows:

$$\tilde{\mathbf{R}}(t) = \frac{\mathbf{R}(t)}{R} = \hat{\mathbf{R}}(t), \tag{12}$$

$$\tilde{\mathbf{V}}(t) = \frac{d\tilde{\mathbf{R}}}{d\tau} = \frac{1}{n_R R}\frac{d\mathbf{R}}{dt} = \frac{\mathbf{V}(t)}{V}. \tag{13}$$

Consequently, $\tilde{\mathbf{R}}(t) = \hat{\mathbf{e}}_r$, $\tilde{\mathbf{V}}(t) = \hat{\mathbf{e}}_f$, and $\tilde{\mu} = 1$, and the values of the following parameters can be found by:

$$\tilde{\mathcal{E}}_m = \frac{1}{2}\tilde{V}^2 - \frac{\tilde{\mu}}{\tilde{R}} = -\frac{1}{2}, \tag{14}$$

$$\tilde{a}_m = \frac{a_m}{R} = 1, \tag{15}$$

$$\tilde{n}_m = \frac{n_R}{n_R} = 1, \tag{16}$$

where the subscript $m$ denotes the mother ship. Moreover, $e_m = 0$, $i_m = i_0$, $\Omega_m = \Omega_0$, and $\omega_m = \omega_0$. Without lost of generality, we set $f_m = 0$ for the mother ship at $t = 0$.

Accordingly, the normalized Gauss equations are given as follows:

$$\frac{d\Omega}{d\tau} = \frac{\tilde{n}\tilde{a}\tilde{r}}{\sqrt{1-e^2}}\tilde{F}_h \sin u \csc i, \tag{17}$$

$$\frac{di}{d\tau} = \frac{\tilde{n}\tilde{a}\tilde{r}}{\sqrt{1-e^2}}\tilde{F}_h \cos u, \tag{18}$$

$$\frac{de}{d\tau} = \tilde{n}\tilde{a}^2\sqrt{1-e^2}\left\{\tilde{F}_r \sin f + \tilde{F}_f(\cos f + \cos E)\right\}, \tag{19}$$

$$\frac{d\omega}{d\tau} = \frac{\tilde{n}\tilde{a}^2}{e}\sqrt{1-e^2}\left\{-\tilde{F}_r \cos f + \tilde{F}_f\left(1 + \frac{\tilde{r}}{\tilde{p}}\sin f\right)\right\} - \cos i\frac{d\Omega}{d\tau}, \tag{20}$$

$$\frac{d\tilde{a}}{d\tau} = 2\tilde{n}\tilde{a}^2\left\{\tilde{F}_r\frac{\tilde{a}e}{\sqrt{1-e^2}}\sin f + \tilde{F}_f\frac{\tilde{a}^2\sqrt{1-e^2}}{\tilde{r}}\right\}, \tag{21}$$

$$\frac{d\tilde{n}}{d\tau} = -\frac{3}{2}\frac{\tilde{n}}{\tilde{a}}\frac{d\tilde{a}}{d\tau}. \tag{22}$$

where $f$ is the true anomaly and $u = f + \omega$.

The remaining part of the paper focuses on the normalized Gauss equations. To simplify future derivations, the tilde signs in Equations (17) to (22) are dropped in the following part of the paper.

Notable, perturbations may cause osculating or non-osculating orbits, depending on the types of perturbations [14,15]. In the problem discussed in this article, orbits are osculating because the PLP thrust is not velocity dependent. Consequently, the term "orbit elements" refers to "osculating orbit elements" in the remainder of this paper.

### 4.2. Review of Perturbation Theory

This paper focused on the influence of small PLP thrust. Hence, perturbation theory can be employed to understand and approximate the variation of the orbit elements. The following is a brief review of perturbation theory.

Consider a differentiable function $x(t)$ that satisfies the following differential equation:

$$\frac{dx}{dt} = \epsilon f(x), \tag{23}$$

where $\epsilon \ll 1$. Suppose the solution to the differential equation is:

$$x(t) = x^{(0)}(t) + \epsilon x^{(1)}(t) + \epsilon^2 x^{(2)}(t) + \mathcal{O}(\epsilon^3), \tag{24}$$

where $x^{(0)}(t)$, $x^{(1)}(t)$, $x^{(2)}(t)$, etc., are assumed differential.

Substituting Equation (24) into Equation (23) yields:

$$
\begin{aligned}
\dot{x}^{(0)} + \epsilon \dot{x}^{(1)} + \epsilon^2 \dot{x}^{(2)} + \cdots &= \epsilon f(x^{(0)} + \epsilon x^{(1)} + \epsilon^2 x^{(2)} + \cdots) \\
&= \epsilon \left\{ f(x^{(0)}) + \left. \frac{\partial f}{\partial \epsilon} \right|_{\epsilon=0} \epsilon + \cdots \right\}.
\end{aligned} \tag{25}
$$

Comparing the coefficients between two sides leads to:

$$\frac{dx^{(0)}}{dt} = 0, \tag{26}$$

$$\frac{d\epsilon x^{(1)}}{dt} = \epsilon f(x^{(0)}), \tag{27}$$

$$\vdots$$

Equation (26) determines a constant $x^{(0)}$, and $\epsilon x^{(1)}$ can be found by:

$$\epsilon x^{(1)} = \int \epsilon f(x^{(0)}) dt. \tag{28}$$

If $\epsilon$ is small enough, the solution $x(t)$ can be approached by the first-order approximation as follows:

$$x(t) \approx x^{(0)} + \epsilon x^{(1)}(t). \tag{29}$$

Note that the notations for the zeroth-order approximation are changed to $\bar{x}$ in the following derivations to make the equations neat, that is,

$$x(t) \approx \bar{x}(t) + \epsilon x^{(1)}(t). \tag{30}$$

## 5. Influence of Planar PLP Thrust

If the PLP force is planar, i.e., $\mathbf{F} = F_r \hat{\mathbf{e}}_r + F_f \hat{\mathbf{e}}_f$, it is apparent that $d\Omega/d\tau = 0$ and $di/d\tau = 0$. The remaining equations to solve are Equations (19)–(22). Notably, $d\Omega/d\tau$ in Equation (20) equals zero.

### 5.1. Zeroth-Order Approximation

The geometric parameters of the mother ship and mission spacecraft are depicted in Figure 4. As determined in Equation (26), the zeroth-order approximation is the unperturbed orbital elements of the mission spacecraft. Here, we assumed that the mission spacecraft deviates from the mother ship by a normalized offset $||\mathbf{L}(0)|| = l_0 \ll 1$, but keeps a velocity identical to the mother ship, i.e., $\mathbf{v} = \mathbf{V} = \mathbf{j}$ at $t = 0$. A potential scenario to implement this assumption is that the mother ship holds the mission spacecraft with robotic arms to place it at the launch site in the vicinity of the mother ship. When the laser beam is initiated, the robotic arms release and the spacecraft is driven away.

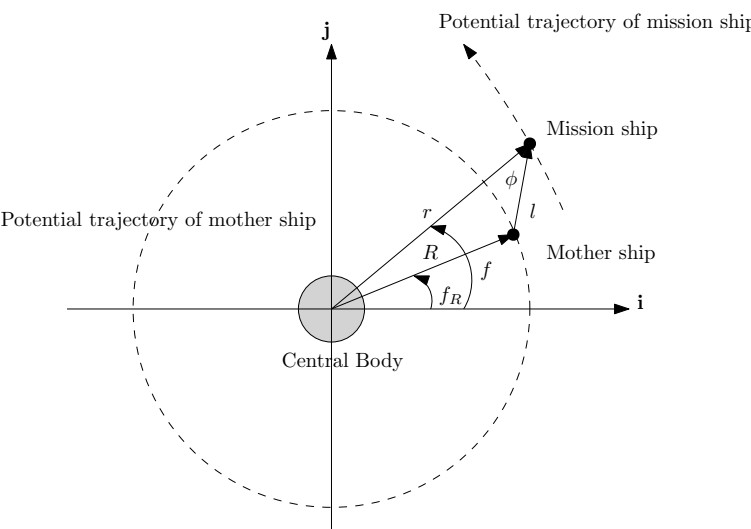

**Figure 4.** Parameters of the mother mother ship and mission spacecraft. All parameters are normalized in this figure.

### 5.1.1. Semi-Major Axis

The zeroth-order approximation of semi-major axis $\bar{a}$ can be obtained from the initial energy. According to the assumption, the initial energy of the mission spacecraft is obtained by:

$$\mathcal{E} = \frac{1}{2} \times 1^2 - \frac{1}{r}. \tag{31}$$

Since $1 - l_0 \leq r \leq 1 + l_0$, it is conclusive that:

$$
\begin{aligned}
\mathcal{E}_{\min} &= \frac{1}{2} - \frac{1}{1 - l_0} = \frac{1}{2} - (1 - l_0)^{-1} \\
&\approx \frac{1}{2} - (1 + l_0) = -\frac{1}{2} - l_0.
\end{aligned} \tag{32}
$$

Similarly, $\mathcal{E}_{\max} \approx -1/2 + l_0$. As a result,

$$
\begin{aligned}
\bar{a}_{\min} &= \frac{-1}{2\mathcal{E}_{\min}} \\
&= -(-1 - 2l_0)^{-1} \approx 1 - 2l_0.
\end{aligned} \tag{33}
$$

Similarly, $\bar{a}_{\max} = 1 + 2l_0$. Consequently, we conclude that $\bar{a} \leq 1 + 2l_0$.

### 5.1.2. Eccentricity

To find the initial eccentricity of the mission spacecraft, we had to compute the initial angular momentum. Let $\psi$ be the angle between $\mathbf{L}$ and $\mathbf{R}$. Figure 4 indicates that $\mathbf{r} = (1 + l_0 \cos \psi)\mathbf{i} + l_0 \sin \psi \mathbf{j}$ and $\mathbf{v} = \mathbf{j}$ and:

$$h \quad = \quad ||\mathbf{r} \times \mathbf{v}|| = 1 + l_0 \cos \psi, \tag{34}$$

and $r = \sqrt{1 + 2l_0 \cos \psi + l_0^2}$. The eccentricity is computed by:

$$
\begin{aligned}
e \quad &= \quad \sqrt{1 + \frac{2\mathcal{E}h^2}{\mu^2}} \\
&= \quad \sqrt{1 + 2\left(\frac{1}{2} - \frac{1}{r}\right)(1 + l_0 \cos \psi)^2} \\
&= \quad \sqrt{1 + \left[1 - 2(1 + 2l_0 \cos \psi + l_0^2)^{(-1/2)}\right](1 + l_0 \cos \psi)^2} \\
&\approx \quad \sqrt{l_0^2 + k_1 l_0^3 + k_2 l_0^4} \\
&\approx \quad l_0,
\end{aligned}
\tag{35}
$$

where $k_1 = 2 \cos \psi - 4 \cos^3 \psi$, $k_2 = \cos^2 \psi - 3 \cos^4 \psi$, and the approximation of $1/r$ is detailed in Appendix A. As a result, $\bar{e} = l_0$.

### 5.1.3. Longitude of the Ascending Node and Argument of Periapsis

Subject to an initial planar offset, only the magnitude of the angular momentum of the mission spacecraft may change, but the axis of the angular momentum will not. This indicates that the longitude of ascension remains identical to the mother ship, i.e., $\bar{\Omega} = \Omega_m$.

However, the initial offset of the mission spacecraft may cause the shift of the eccentricity axis. Since the argument of periapsis is defined as the angle between the eccentricity axis and the axis of the ascending node, the zeroth-order of the argument of periapsis may change. An easier approach is to find the angle difference from the mother ship, i.e., $\bar{\omega} = \omega_m + \Delta\omega$, where $\Delta\omega$ satisfies:

$$\mathbf{e} \cdot \mathbf{R} \quad = \quad e \cos \Delta\omega. \tag{36}$$

Recall that $\mathbf{r} = (1 + l_0 \cos \psi))\mathbf{i} + l_0 \sin \psi \mathbf{j}$ and $\mathbf{h} = \mathbf{r} \times \mathbf{v} = (1 + l_0 \cos \psi))\mathbf{k}$. Then,

$$
\begin{aligned}
\mathbf{e} \quad &= \quad (\mathbf{v} \times \mathbf{h})/\mu - \hat{\mathbf{r}} \\
&= \quad (1 + l_0 \cos \psi)\mathbf{i} - \frac{(1 + l_0 \cos \psi)\mathbf{i} + l_0 \sin \psi \mathbf{j}}{\sqrt{1 + l_0^2 + 2l_0 \cos \psi}}.
\end{aligned}
\tag{37}
$$

Consequently,

$$
\begin{aligned}
\mathbf{e} \cdot \mathbf{R} \quad &= \quad (1 + l_0 \cos \psi) - \frac{1 + l_0 \cos \psi}{\sqrt{1 + l_0^2 + 2l_0 \cos \psi}} \\
&= \quad (1 + l_0 \cos \psi)\left(1 - \frac{1}{\sqrt{1 + l_0^2 + 2l_0 \cos \psi}}\right) \\
&\approx \quad 2l_0 \cos \psi (1 + l_0 \cos \psi) \approx 2l_0 \cos \psi.
\end{aligned}
\tag{38}
$$

Since $\bar{e} \approx l_0$, we conclude that:

$$\Delta\omega \quad = \quad \cos^{-1}(2 \cos \psi). \tag{39}$$

### 5.1.4. Mean Motion

The mean motion is a function of the semi-major axis, given by $n = \sqrt{\mu/a^3}$. In the normalized system,

$$n = \sqrt{\frac{1}{a^3}} = a^{-\frac{3}{2}} \tag{40}$$

Then, the zeroth-order approximation can be found by:

$$\begin{aligned} \bar{n} &= \bar{a}^{-\frac{3}{2}} = (1 + \Delta a)^{-\frac{3}{2}} \\ &\approx 1 - \frac{3}{2}\Delta a \end{aligned} \tag{41}$$

Since $\bar{a} \in [1 - 2l_0, 1 + 2l_0]$, we conclude that:

$$1 - 3l_0 \le \bar{n} \le 1 + 3l_0 \tag{42}$$

### 5.2. First-Order Approximation and Bounds of Elements under Small Thrust

Suppose the PLP force is small enough so that the right-hand-side functions in Equations (17) to (22) are very small. The perturbation theory can be applied to approximate the solutions. We only consider the influence on planar motion in this section.

### 5.2.1. Semi-Major Axis

According to the conclusion in Section 4.2, the evolution of the semi-major axis can be approximated as:

$$a \approx \bar{a} + \epsilon a^{(1)} + \cdots, \tag{43}$$

where $\bar{a}$ is the solution of the unperturbed system. Section 5.1.1 concluded that $\bar{a} \in [1 - 2l_0, 1 + 2l_0]$, depending on the initial offset of the mission spacecraft.

The first-order approximation $\epsilon a^{(1)}$ can be found by:

$$\begin{aligned} \epsilon a^{(1)} &= \int \epsilon f(\bar{a}, \bar{e}, \bar{n}) d\tau \\ &= \int 2\bar{n}\bar{a}^2 \left\{ F_r \frac{\bar{a}\bar{e}}{\sqrt{1 - \bar{e}^2}} \sin f + F_f \frac{\bar{a}^2\sqrt{1 - \bar{e}^2}}{r} \right\} d\tau \end{aligned} \tag{44}$$

The integration in the time domain is very complicated. Instead, we integrated this system in the eccentric-anomaly domain. According to Kepler's law, $n\tau = E - e \sin E$. We can change the variables by:

$$n d\tau = (1 - e \cos E) dE \tag{45}$$

Moreover, the following identities were employed:

$$r = \frac{a(1 - e^2)}{1 + e \cos f} = a(1 - e \cos E) \tag{46}$$

$$\cos f = \frac{\cos E - e}{1 - e \cos E} \tag{47}$$

$$\sin f = \frac{\sqrt{1 - e^2} \sin E}{1 - e \cos E} \tag{48}$$

Then, the integration can be rewritten as:

$$\epsilon a^{(1)} = \int_0^E 2\bar{a}^3 \left( F_r \bar{e} \sin E + F_f \sqrt{1 - \bar{e}^2} \right) dE, \tag{49}$$

where $E \geq 0$. The assumption of $E \geq 0$ holds for the whole article.

As shown in Figure 4, $F_r = F \cos \phi$ and $F_f = F \sin \phi$, where $\phi$ satisfies:

$$l^2 + r^2 - 2lr \cos \phi \;=\; 1 \tag{50}$$

Moreover,

$$\frac{1}{\sin \phi} \;=\; \frac{l}{\sin(f - \tau)} \tag{51}$$

$$r \;=\; \frac{a(1 - e^2)}{1 + e \cos f} \tag{52}$$

Solving Equations (50) to (52) yields $\phi$ as a function of $f$, i.e., $\phi = \phi(f)$. Since the true anomaly $f$ is a function of $E$, described in Equations (47) and (48), by the change of variables, we eventually obtained $\phi = \phi(E)$.

The integration can be written as:

$$\epsilon a^{(1)} \;=\; \int_0^E 2\bar{a}^3 F\left(\bar{e} \cos \phi \sin E + \sin \phi \sqrt{1 - \bar{e}^2}\right) dE. \tag{53}$$

Define:

$$\cos \alpha_a \;=\; \frac{\bar{e} \sin E}{\sqrt{\bar{e}^2 \sin^2 E + 1 - \bar{e}^2}} = \frac{\bar{e} \sin E}{\sqrt{1 - \bar{e}^2 \cos^2 E}} \tag{54}$$

$$\sin \alpha_a \;=\; \frac{\sqrt{1 - \bar{e}^2}}{\sqrt{1 - \bar{e}^2 \cos^2 E}} \tag{55}$$

Then, Equation (53) can be rewritten as:

$$\epsilon a^{(1)} \;=\; \int_0^E 2\bar{a}^3 F \sqrt{1 - \bar{e}^2 \cos^2 E} \cos(\phi - \alpha_a) dE \tag{56}$$

Since $\sqrt{1 - \bar{e}^2 \cos^2 E} \leq 1$ and $-1 \leq \cos(\phi - \alpha_a) \leq 1$, the integration satisfies,

$$
\begin{aligned}
\epsilon a^{(1)} &= \int_0^E 2\bar{a}^3 F \sqrt{1 - \bar{e}^2 \cos^2 E} \cos(\phi - \alpha_a) dE \\
&\leq \int_0^E 2\bar{a}^3 F dE \\
&= 2EF\bar{a}^3
\end{aligned}
\tag{57}
$$

If we integrate over one period, i.e., $E = 2\pi$, then $\epsilon a^{(1)} \leq 4\pi F \bar{a}^3$. Accordingly,

$$
\begin{aligned}
a &\leq 1 \pm 2l_0 + 4\pi F(1 \pm 2l_0)^3 \\
&\approx 1 \pm 2l_0 + 4\pi F(1 \pm 6l_0) \\
&= (1 + 4\pi F) \pm (2 \pm 24\pi F)l_0
\end{aligned}
\tag{58}
$$

However, Equation (58) holds only if the PLP force is small enough. In other words, $\epsilon a^{(1)} \ll \bar{a}$. A reasonable assumption is that:

$$\epsilon a^{(1)} \leq 4\pi F \bar{a}^3 \leq 0.1\bar{a} \tag{59}$$

Then, we conclude:

$$
\begin{aligned}
F &\leq \frac{1}{40\pi \bar{a}^2} \\
&\approx \frac{1}{40\pi}(1 \mp 4l_0) \approx 0.008
\end{aligned}
\tag{60}
$$

Equation (60) provides a sufficient condition. If $F$ is smaller than 0.008, the trajectory will not leave the vicinity of the mother ship for a while. Actually, approximations and assumptions were made during the derivation. The upper bound of $F$ for a small perturbation can be larger. The result also agrees with the upper bound found in Hsiao's paper [9], where a more rigorous bound was applied by $F \le l_0/16$.

5.2.2. Eccentricity

A similar procedure can be applied to analyze the evolution of eccentricity. With the change of variables, Equation (19) can be rewritten as:

$$\epsilon e^{(1)} = \int_0^E \bar{a}^2 \sqrt{1-\bar{e}^2}\left(F_r\sqrt{1-\bar{e}^2}\sin E + F_f(2\cos E - \bar{e} - \bar{e}\cos^2 E)\right)dE \tag{61}$$

Defining $K_e = \sqrt{1-\bar{e}^2}$ and applying the identity $\cos^2 E = (1+\cos 2E)/2$ yield:

$$\begin{aligned}
\epsilon e^{(1)} &= \int_0^E \bar{a}^2 K_e\left(F_r K_e \sin E + F_f\left(2\cos E - \frac{3\bar{e}}{2} - \frac{\bar{e}\cos 2E}{2}\right)\right)dE \\
&= \int_0^E \bar{a}^2 F K_e\left(\cos\phi K_e \sin E + \sin\phi\left(2\cos E - \frac{3\bar{e}}{2} - \frac{\bar{e}\cos 2E}{2}\right)\right)dE
\end{aligned} \tag{62}$$

Define

$$\begin{aligned}
K &= (K_e \sin E)^2 + \left(2\cos E - \frac{3\bar{e}}{2} - \frac{\bar{e}\cos 2E}{2}\right)^2 \\
&= \bar{e}^2\cos^4 E - 4\bar{e}\cos^3 E + (-\bar{e}^4 + 4\bar{e}^2 + 3)\cos^2 E \\
&\quad -4\bar{e}\cos E + (-\bar{e}^2 + 1 + \bar{e}^4) \ge 0
\end{aligned} \tag{63}$$

$$\cos\alpha_e = \frac{K_e \sin E}{\sqrt{K}} \tag{64}$$

$$\sin\alpha_e = \frac{2\cos E - 3\bar{e}/2 - \bar{e}\cos 2E/2}{\sqrt{K}} \tag{65}$$

Equation (62) becomes:

$$\epsilon e^{(1)} = \int_0^E \bar{a}^2 F K_e \sqrt{K}\cos(\phi - \alpha_e)dE \tag{66}$$

The parameter $K$ in Equation (63) is a function of $\bar{e}$ and $\cos E$. The mission spacecraft is assumed to launch very close to the mother ship. It is reasonable to assume $0 \le \bar{e} \ll 1$. Moreover, $-1 \le \cos E \le 1$.

Figure 5 gives the simulation of the maximum value versus different $\bar{e}$'s. We can see that, over all possible $\bar{e}$'s, $\sqrt{K} \le 2$. As a result, we conclude that Equation (62) leads to:

$$\begin{aligned}
\epsilon e^{(1)} &= \int_0^E \bar{a}^2 F K_e \sqrt{K}\cos(\phi - \alpha_e)dE \\
&\le \int_0^E 2\bar{a}^2 F K_e \cos(\phi - \alpha_e)dE \\
&\le 2E\bar{a}^2 F K_e
\end{aligned} \tag{67}$$

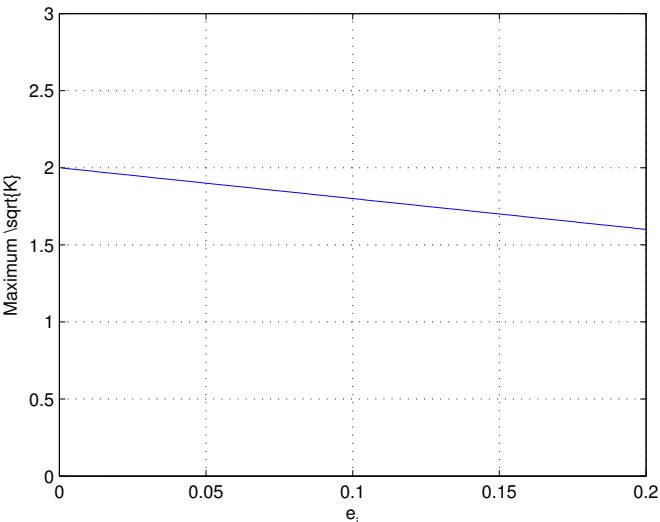

**Figure 5.** The maximum value of $\sqrt{K}$ over $E$ in Equation (63) as a function of eccentricity $\bar{e}$.

If the system is integrated over one period, i.e., $E = 2\pi$, we obtain:

$$\epsilon e^{(1)} \quad \leq \quad 4\pi \bar{a}^2 F \sqrt{1 - \bar{e}^2} \tag{68}$$

To satisfy the assumption of small perturbation, we apply $F \leq (1/40\pi)$ and obtain:

$$\epsilon e^{(1)} \quad \leq \quad \frac{\bar{a}^2 \sqrt{1 - \bar{e}^2}}{10} \tag{69}$$

Thus,

$$
\begin{aligned}
e \quad &\leq \quad \bar{e} + \frac{\bar{a}^2 \sqrt{1 - \bar{e}^2}}{10} \\
&\approx \quad l_0 + \frac{\bar{a}^2}{10}
\end{aligned}
\tag{70}
$$

Since $1 - 2l_0 \leq \bar{a} \leq 1 + 2l_0$, the upper bound of $e$ varies between:

$$
\begin{aligned}
l_0 + \frac{1}{10}(1 \mp 2l_0)^2 \quad &\approx \quad l_0 + \frac{1}{10}(1 \mp 4l_0) \\
&= \quad 0.1 + 0.6l_0 \quad \text{or} \quad 0.1 + 1.4l_0
\end{aligned}
\tag{71}
$$

As a result, we conclude that $e \leq 0.1 + 1.4l_0$.

### 5.2.3. Argument of Periapsis

The first-order approximation of the argument of periapsis is governed by Equation (20). In this case, only a planar PLP force is applied. It is conclusive that $d\Omega/d\tau = 0$, implying that $F_r$ and $F_f$ are the only two terms that influence the variation of periapsis. The first-order approximation is attainable by integrating Equation (20), given by:

$$
\begin{aligned}
\epsilon \omega^{(1)} \quad &= \quad \int_0^{2\pi} \frac{\bar{n}\bar{a}^2}{\bar{e}} \sqrt{1 - \bar{e}^2} \left\{ -\bar{F}_r \cos f + \bar{F}_f \left( 1 + \frac{r}{p} \sin f \right) \right\} d\tau \\
&= \quad \int_0^{2\pi} \frac{\bar{a}^3 K_e}{\bar{e}} \left\{ -\bar{F}_r (\cos E - e) + \bar{F}_f \left( 1 + \frac{1 - e \cos E}{K_e} \sin E \right) \right\} dE
\end{aligned}
\tag{72}
$$

Recall that $F_r = F \cos \phi$ and $F_f = F \sin \phi$. The integral can be further written as:

$$\epsilon \omega^{(1)} = \frac{\bar{a}^3 F}{\bar{e}} \int_0^{2\pi} \{K_1(\bar{e}, E) \cos \phi + K_2(\bar{e}, E) \sin \phi\} dE \tag{73}$$

where $K_1(\bar{e}, E) = -K_e(\cos E - \bar{e})$ and $K_2(\bar{e}, E) = K_e + \sin E - \bar{e} \sin E \cos E$. Define $K_\omega = K_1^2 + K_2^2$, and:

$$\sin \alpha_\omega = \frac{K_1}{K_\omega} \tag{74}$$

$$\cos \alpha_\omega = \frac{K_2}{K_\omega} \tag{75}$$

The integration can be shortened as:

$$\begin{aligned} \epsilon \omega^{(1)} &= \frac{\bar{a}^3 F}{\bar{e}} \int_0^{2\pi} \sqrt{K_\omega} \sin(\phi + \alpha_\omega) dE \\ &\leq \frac{\bar{a}^3 F}{\bar{e}} \int_0^{2\pi} \sqrt{K_\omega} dE, \end{aligned} \tag{76}$$

where $K_\omega$ is a function of $\bar{e}$ and $E$. Because $K_\omega$ is highly nonlinear, it is difficult to integrate Equation (76) analytically. Let $K_{\omega\text{max}}(\bar{e})$ denote the maximum value of $K_\omega(\bar{e}, E)$ over $E = [0, 2\pi]$ given $\bar{e}$. Then, the upper bound of Equation (76) can be found by:

$$\epsilon \omega^{(1)} \leq 2\pi \sqrt{K_{\omega\text{max}}(\bar{e})} \frac{\bar{a}^3 F}{\bar{e}}. \tag{77}$$

The value of $\sqrt{K_{\omega\text{max}}(\bar{e})}$ for a given $\bar{e}$ is presented in Figure 6 numerically. One can see that $\sqrt{K_\omega(\bar{e}, E)} \leq 2.1343$ for all potential eccentricities. Since $\bar{e} = l_0$, it is reasonable to assume that $\bar{e} \leq 0.1$ and $\sqrt{K_\omega} \leq 2.02$ according to Figure 6. Moreover, to satisfy the assumption of small perturbation, we let $F \leq 0.008$, or equivalently, $4\pi\bar{a}^3 F \leq 0.1\bar{a}$, as in our previous discussion. This assumption leads to $2\pi\bar{a}^3 F \leq 0.05\bar{a}$, and the first-order approximation can be concluded as:

$$\begin{aligned} \epsilon \omega^{(1)} &\leq 2.02 \cdot \frac{2\pi\bar{a}^3 F}{\bar{e}} \\ &= 0.1\frac{\bar{a}}{l_0} \\ &= 0.1\frac{1 \pm 2l_0}{l_0} \\ &\approx \frac{0.1}{l_0} \pm 0.2 \end{aligned} \tag{78}$$

with $l_0 \leq 0.1$. Note that this result may have a large error for $l_0 \approx 0$.

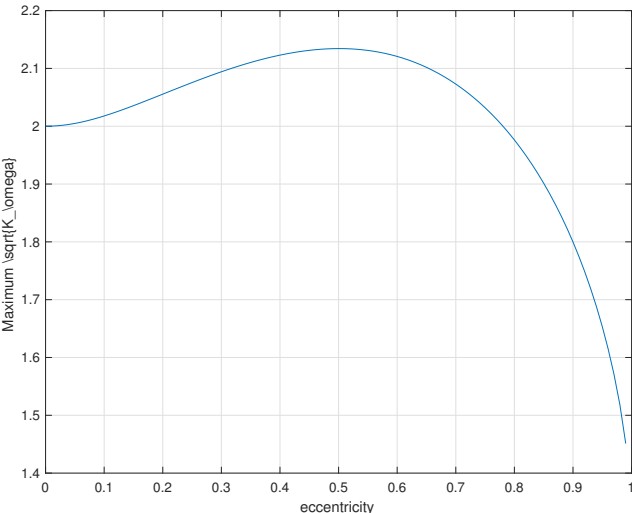

**Figure 6.** The maximum value of $\sqrt{K_\omega}$ over $E$ as a function of eccentricity $\bar{e}$.

5.2.4. Mean Motion

The evolution of mean motion is described in Equation (22). Similarly, under small perturbations, the mean motion can be approximated by $n \approx \bar{n} + \epsilon n^{(1)}$ with $(a, e)$ substituted by their zeroth-order approximations.

By handingthe integration variable to $E$, Equation (22) is rewritten as:

$$
\begin{aligned}
\frac{d\epsilon n^{(1)}}{dE} &= -\frac{3}{2}\frac{\bar{n}}{\bar{a}}\frac{da}{dE} \\
&= -\frac{3}{2}\frac{\bar{n}}{\bar{a}}\frac{d(\bar{a} + \epsilon a^{(1)})}{dE} \\
&= -\frac{3}{2}\frac{\bar{n}}{\bar{a}}\frac{d\epsilon a^{(1)}}{dE}
\end{aligned}
\tag{79}
$$

Integrating the two sides over $2\pi$, with the result of Equation (57), yields:

$$
\begin{aligned}
\epsilon n^{(1)} &= -\frac{3}{2}\frac{\bar{n}}{\bar{a}}\int \frac{d\epsilon a^{(1)}}{dE}dE \\
&\geq -\frac{3}{2}\frac{\bar{n}}{\bar{a}}(4\pi F\bar{a}^3)
\end{aligned}
\tag{80}
$$

The lower bound of $\epsilon n^{(1)}$ depends on the value of $4\pi F\bar{a}^3$. According to Equation (59), we may let $4\pi F\bar{a}^3 = K_n\bar{a}$ with $0 \leq K_n \leq 0.1$. Plugging the result into Equation (80) yields:

$$
\epsilon n^{(1)} \geq -\frac{3K_n}{2}\bar{n}
\tag{81}
$$

Accordingly, the approximation of $n$ is obtained by:

$$
\begin{aligned}
n &= \bar{n} + \epsilon n^{(1)} \\
&\geq \left(1 - \frac{3K_n}{2}\right)\bar{n}
\end{aligned}
\tag{82}
$$

Moreover, $1 - 3l_0 \leq \bar{n} \leq 1 + 3l_0$, and the lower bound of the mean motion varies between:

$$
\left(1 - \frac{3K_n}{2}\right)(1 \mp 3l_0) \approx 1 - \frac{3K_n}{2} \mp 3l_0
\tag{83}
$$

As a result, we conclude that $n \geq 1 - \frac{3K_n}{2} - 3l_0$ with $0 \leq K_n \leq 0.1$.

## 6. Influence of Out-of-Plane PLP Thrust

The out-of-plane PLP thrust applies along the direction perpendicular to the orbital plane of the the mother ship initially. All parameters are shown in Figure 7. The initial position offset is assumed $\mathbf{L}(0) = l_0\mathbf{k}$. Hence, the position and velocity of the mission spacecraft are, respectively, given by $\mathbf{r} = \mathbf{i} + l_0\mathbf{k}$ and $\mathbf{v} = \mathbf{j}$.

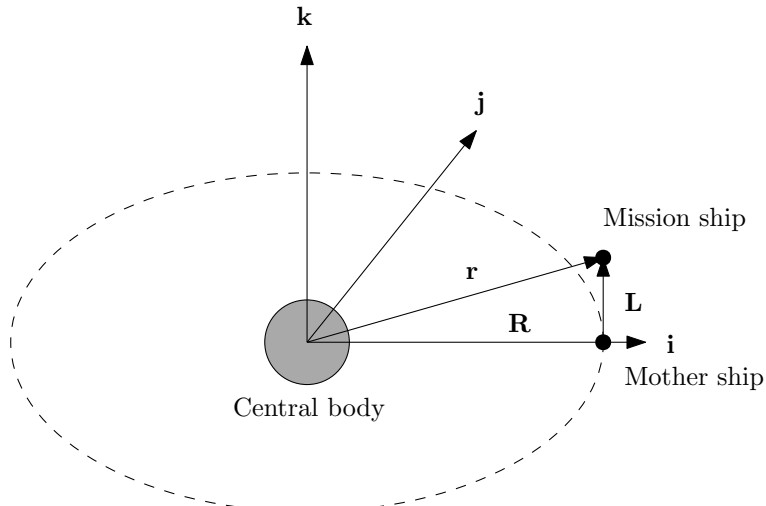

**Figure 7.** Parameters of the mother ship and mission spacecraft subject to out-of-plane PLP thrust. All parameters are normalized in this figure.

Although the out-of-plane cases were studied in this paper, in practice, a pure out-of-plane PLP thrust for the whole mission is infeasible. When the mission spacecraft is initially driven by an out-of-plane PLP thrust, in order to aim at the mission spacecraft, the motion of the spacecraft will naturally cause planar components of the PLP force. As a result, a practical motion subject to initial out-of-plane PLP thrust will eventually be described by the combination of planar and out-of-plane effects.

### 6.1. Zeroth-Order Approximation

6.1.1. Semi-Major Axis, Eccentricity, and Mean Motion

According to Figure 7, it can be found that:

$$r = \sqrt{1 + l_0^2} \approx 1 \tag{84}$$

$$\mathbf{h} = \mathbf{r} \times \mathbf{v} = -l_0\mathbf{i} + \mathbf{k}. \tag{85}$$

Consequently, $h = ||\mathbf{h}|| = \sqrt{1 + l_0^2} \approx 1$. Moreover, the value of the mission ship energy, with $\mu = 1$, is computed as:

$$\mathcal{E} = \frac{1}{2} \times v^2 - \frac{\mu}{r} = -\frac{1}{2} \tag{86}$$

The approximated semi-major axis is:

$$\bar{a} = -\frac{\mu}{2\mathcal{E}} = 1 \tag{87}$$

The approximated eccentricity is:

$$\bar{e} = \sqrt{1 + \frac{2\mathcal{E}h^2}{\mu^2}} = 0 \tag{88}$$

The mean motion $\bar{n}$ is a function of semi-major axis $\bar{a}$. Since the semi-major axis does not change in the zeroth-order approximation, the mean motion remains unchanged as well, i.e., $\bar{n} = 1$.

### 6.1.2. Inclination

The inclination of the mission spacecraft is influenced by the orbital elements of the mother ship. Let $i_m$, $\omega_m$, and $\Omega_m$ denote the inclination, argument of periapsis, and longitude of the ascending node of the mother ship, respectively. The transformation of the orbit reference to the J2000 coordinate system is given by [13]:

$$\mathbf{p}_J \quad = \quad R_x(-f_m)R_z(-\Omega_m)R_x(-i_m)R_z(-\omega_m)\mathbf{p}_{OR} \tag{89}$$

where $\mathbf{p}_{OR} = x\mathbf{i} + y\mathbf{j} + z\mathbf{k}$ is a vector with respect to the orbital reference and $\mathbf{p}_J = X\mathbf{I} + Y\mathbf{J} + Z\mathbf{K}$ is with respect to the J2000 system. Moreover, $R_x(\theta)$ and $R_z(\theta)$ are the rotation matrices with respect to the $\mathbf{i}$- and $\mathbf{k}$-axis, respectively.

Recall that $\mathbf{h} = -l_0\mathbf{i} + \mathbf{k}$. Therefore,

$$\begin{aligned}
\cos\bar{i} \quad &= \quad \frac{\mathbf{h}\cdot\mathbf{K}}{||\mathbf{h}||} \\
&= \quad \frac{1}{\sqrt{1+l_0^2}}(\cos i_m - l_0\sin i_m \sin\omega_m)
\end{aligned} \tag{90}$$

with $f_m$ assumed to be zero. The inclination of mission spacecraft can be found by:

$$\begin{aligned}
\bar{i} \quad &= \quad \cos^{-1}\left(\frac{\cos i_m - l_0\sin i_m \sin\omega_m}{\sqrt{1+l_0^2}}\right) \\
&\approx \quad \cos^{-1}(\cos i_m) + \left.\frac{d\cos^{-1}(\chi)}{d\chi}\frac{d\chi}{dl_0}\right|_{l_0=0}l_0 \\
&= \quad i_m - \frac{1}{\sqrt{1-\chi^2}}(-l_0\sin i_m\sin\omega_m) \\
&\approx \quad i_m + \frac{\sin i_m\sin\omega_m}{|\sin i_m|}l_0
\end{aligned} \tag{91}$$

where:

$$\chi \quad = \quad \frac{\cos i_m - l_0\sin i_m\sin\omega_m}{\sqrt{1+l_0^2}} \tag{92}$$

Note that Equation (91) is applicable only if $i_m \neq 0$. Otherwise, this function is singular. In the case that $i_m = 0$, we have to approach the problem in a different way. When $i_m = 0$, Equation (90) degeneratesto:

$$\cos\bar{i} \quad = \quad \frac{1}{\sqrt{1+l_0^2}} \tag{93}$$

The Taylor expansion of $\cos\bar{i}$ is:

$$\cos\bar{i} \quad = \quad 1 - \frac{\bar{i}^2}{2!} + \cdots \tag{94}$$

The second-order approximation of the right-hand side gives:

$$
\frac{1}{\sqrt{1 + l_0^2}} = (1 + l_0^2)^{-1/2}
$$

$$
= 1 - \frac{l_0^2}{2!} + \cdots \tag{95}
$$

Equating the two sides yields $\bar{i} \approx l_0$.

6.1.3. Longitude of Ascending Node and Argument of Periapsis

Recall that the angular momentum of the mission spacecraft is $\mathbf{h} = -l_0\mathbf{i} + \mathbf{k}$. As a result, the ascending node $\mathbf{n}$ can be found by:

$$
\mathbf{n} = \frac{\mathbf{K} \times \mathbf{h}}{||\mathbf{K} \times \mathbf{h}||}
$$

$$
= \frac{n_x\mathbf{I} + n_y\mathbf{J}}{\sqrt{n_x^2 + n_y^2}}, \tag{96}
$$

where:

$$
n_x = l_0(\sin\Omega_m \cos\omega_m + \cos\Omega_m \cos i_m \sin\omega_m) + \cos\Omega_m \sin i_m \tag{97}
$$

$$
n_y = -l_0(\cos\Omega_m \cos\omega_m - \sin\Omega_m \cos i_m \sin\omega_m) + \sin\Omega_m \sin i_m \tag{98}
$$

As a result, the zeroth-order approximation of the longitude of ascending node is obtained by:

$$
\bar{\Omega} = \tan^{-1}\left(\frac{n_y}{n_x}\right) \tag{99}
$$

Given $l_0 = 0$, $\Omega = \Omega_m$.

The transverse of the node, denoted as $\mathbf{n}_\perp$, can be found by:

$$
\mathbf{n}_\perp = \frac{\mathbf{h} \times \mathbf{n}}{||\mathbf{h} \times \mathbf{n}||}
$$

$$
= \frac{n_{\perp_x}\mathbf{I} + n_{\perp_y}\mathbf{J} + n_{\perp_z}\mathbf{K}}{\sqrt{n_{\perp_x}^2 + n_{\perp_y}^2 + n_{\perp_z}^2}} \tag{100}
$$

where:

$$
n_{\perp_x} = -\sqrt{1 + l_0^2}\cos\bar{i}(\sin\Omega_m \sin i_m - l_0 \cos\Omega_m \cos\omega_m +
$$
$$
l_0 \sin\Omega_m \cos i_m \sin\omega_m) \tag{101}
$$

$$
n_{\perp_y} = \sqrt{1 + l_0^2}\cos\bar{i}(\cos\Omega_m \sin i_m + l_0 \sin\Omega_m \cos\omega_m +
$$
$$
l_0 \cos\Omega_m \cos i_m \sin\omega_m) \tag{102}
$$

$$
n_{\perp_z} = l_0^2(\cos^2 i_m - \cos^2 i_m \cos^2 \omega_m + \cos^2 \omega_m) +
$$
$$
2l_0 \cos i_m \sin i_m \sin\omega_m + \sin^2 i_m \tag{103}
$$

and the quantity $\sqrt{1 + l_0^2}\cos\bar{i}$ is presented in Equation (90). Consequently,

$$
\tan\bar{\omega} = \frac{\mathbf{r} \cdot \mathbf{n}_\perp}{\mathbf{r} \cdot \mathbf{n}}
$$

$$
= \frac{\sin i_m \sin\omega_m + l_0 \cos i_m}{\cos\omega_m \sin i_m} \tag{104}
$$

Given $l_0 = 0$, $\omega = \omega_m$.

*6.2. First-Order Approximation*

6.2.1. Inclination

The variation of inclination subject to a small continuous thrust is described in Equation (18). Consider an out-of-plane, small PLP thrust $F$ acting on the mission ship, i.e., $F_h = F$. Integrating Equation (18) over $2\pi$:

$$\epsilon i^{(1)} \quad = \quad \int_0^{2\pi} \frac{\bar{n}\bar{a}\bar{r}}{\sqrt{1 - \bar{e}^2}} F \cos(f + \bar{\omega}) d\tau \tag{105}$$

The trigonometric function in the integrant can be further expanded using the identity $\cos(f + \bar{\omega}) = \cos f \cos \bar{\omega} - \sin f \sin \bar{\omega}$. Along with the change of variables in Equations (45) to (48), the integral can be written as:

$$
\begin{aligned}
\epsilon i^{(1)} \quad &= \quad \int_0^{2\pi} \frac{\bar{a}\bar{r}}{\sqrt{1 - \bar{e}^2}} F \Big( (\cos E - \bar{e}) \cos \bar{\omega} - \sqrt{1 - \bar{e}^2} \sin E \sin \bar{\omega} \Big) dE \\
&= \quad -\frac{\bar{a}^2 \bar{e} F \cos \bar{\omega}}{\sqrt{1 - \bar{e}^2}} \int_0^{2\pi} \cos^2 E dE \\
&= \quad -\frac{\pi \bar{a}^2 \bar{e} F \cos \bar{\omega}}{\sqrt{1 - \bar{e}^2}} = 0
\end{aligned}
\tag{106}
$$

since $\bar{e} = 0$.

6.2.2. Longitude of Ascending Node

The variation of inclination subject to a small continuous thrust is described in Equation (18). Consider an out-of-plane, small PLP thrust $F$ acting on the mission ship, i.e., $F_h = F$. Integrating Equation (18) over $2\pi$:

$$\epsilon \Omega^{(1)} \quad = \quad \int_0^{2\pi} \left( \frac{\bar{n}\bar{a}\bar{r}}{\sqrt{1 - \bar{e}^2}} F \sin(f + \bar{\omega}) \csc \bar{i} \right) d\tau \tag{107}$$

Similarly, the trigonometric identity and the change of variable bring:

$$
\begin{aligned}
\epsilon \Omega^{(1)} \quad &= \quad \int_0^{2\pi} \frac{\bar{a}\bar{r}}{\sqrt{1 - \bar{e}^2}} F \Big( \sqrt{1 - \bar{e}^2} \sin E \cos \bar{\omega} + (\cos E - \bar{e}) \sin \bar{\omega} \Big) \csc \bar{i} dE \\
&= \quad -\frac{\bar{a}^2 \bar{e} F \sin \bar{\omega} \csc \bar{i}}{\sqrt{1 - \bar{e}^2}} \int_0^{2\pi} \cos^2 E dE \\
&= \quad -\frac{\pi \bar{a}^2 \bar{e} F \sin \bar{\omega} \csc \bar{i}}{\sqrt{1 - \bar{e}^2}} = 0
\end{aligned}
\tag{108}
$$

6.2.3. Argument of Periapsis

The variation of inclination subject to a small continuous thrust is described in Equation (20). With an out-of-plane, small PLP force $F$ acting on the mission ship, i.e., $F_h = F$, as the only thrust, the differential equation of the first-order term degenerates to:

$$
\begin{aligned}
\frac{d\epsilon \omega^{(1)}}{d\tau} \quad &= \quad -\cos \bar{i} \frac{d\epsilon \Omega^{(1)}}{d\tau} \\
&= \quad \frac{\bar{n}\bar{a}\bar{r}}{\sqrt{1 - \bar{e}^2}} F \sin(f + \bar{\omega}) \cot \bar{i}
\end{aligned}
\tag{109}
$$

It is obvious that the integral of Equation (109) over $2\pi$ is:

$$
\begin{aligned}
\epsilon \omega^{(1)} &= \int_0^{2\pi} \left( \frac{\bar{n}\bar{a}\bar{r}}{\sqrt{1-\bar{e}^2}} F \sin(f + \bar{\omega}) \cot \bar{i} \right) d\tau \\
&= \frac{\pi \bar{a}^2 \bar{e} F \sin \bar{\omega} \cot \bar{i}}{\sqrt{1-\bar{e}^2}} = 0
\end{aligned}
\tag{110}
$$

## 7. Numerical Simulations

Figures 8–24 present numerical simulations of several selected cases. The numerical simulations were performed to verify the algorithms and the results proposed in the proceeding sections. Five types of simulations are presented here.

First, a planar PLP thrust was applied to the mission spacecraft for an orbit period. Since the original launching direction may be arbitrary in real missions, we present four examples in this part, corresponding to four initial directions. Secondly, one of the aforementioned four cases was selected to integrate over a longer time period. This simulation was performed to qualitatively demonstrate that a PLP thrust smaller than the proposed criteria is not powerful enough to expel a mission spacecraft for interplanetary travel, as concluded in our algorithm. Thirdly, an out-of-plane example is presented to qualitatively show the variation of the orbit elements. However, the results derived in the preceding section cannot be applied to determine the bound of variation, because the motions in the out-of-plane cases are always coupled with planar components. Fourthly, a dimensional case is presented to show how our results can be applied. Finally, an interesting example shows that a PLP thrust can control a spacecraft formation by the Clohessy–Wiltshire (C-W) equations.

Throughout all the numerical simulations, the magnitude of the normalized PLP thrust was set as $F = 0.008$. The normalized orbit elements of the mother ship were randomly given as $(a_m, e_m, i_m, \Omega_m, \omega_m) = (1, 0, 30°, 40°, 50°)$. Since the mother ship is in a circular orbit, without lost of generality, the initial true anomaly can be set as zero, i.e., $f_m(0) = 0°$. All trajectories that examine the variations of the orbit elements were obtained by integrating Equations (4) and (5) directly and then converted to the osculating elements.

### 7.1. Verification of Bounds with a Planar PLP Thrust

In this section, the PLP force was assumed to be planar. The initial normalized offsets of the mission spacecraft from the mother ship were given as $\mathbf{L}_0 = 0.1\ \mathbf{i}$, $0.1\ \mathbf{j}$, $-0.1\ \mathbf{i}$, and $-0.1\ \mathbf{j}$, respectively.

Figures 8–15 present the simulation results. All the bounds for orbit elements are discussed in Section 5. In Figures 8 and 9, the initial excursion was set as $\mathbf{L}_0 = 0.1\ \mathbf{i}$. The PLP force always aims along the line of sight of the mother ship and mission spacecraft. In Figure 8, the trajectories of mother ship and mission spacecraft are presented with respect to the inertial frame, respectively. In the right-most plot, the relative trajectory of the mission spacecraft to the mother ship is provided, and it is also presented in the inertial frame. Figure 9 shows the variation of orbit elements of the mission spacecraft. Since the true anomaly has nothing to do with the variation of the orbit, it is not presented in the figure. Every bound over one normalized orbit period is provided in the title of the corresponded subplot and also presented as a dashed line. It was verified that the bounds were well predicted by our algorithm. Similar results can also be observed in Figures 10–15, corresponding to different initial conditions.

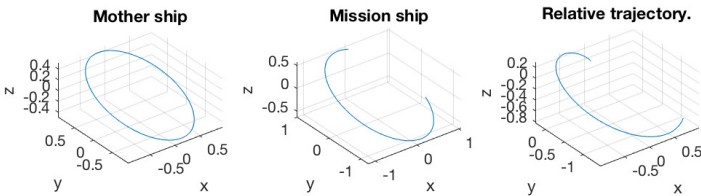

**Figure 8.** (**Left**) Trajectory of the mother ship; (**middle**) trajectory of the mission spacecraft; (**right**) relative trajectory of the mission spacecraft to the mother ship. The initial excursion is $\mathbf{L}_0 = 0.1\,\mathbf{i}$.

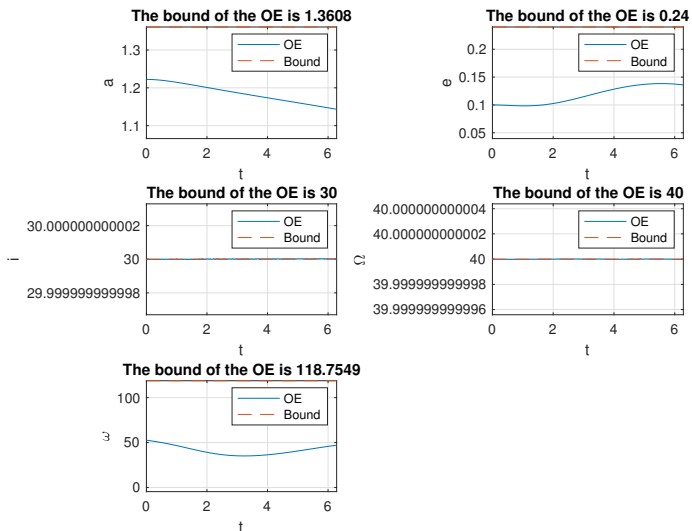

**Figure 9.** Variation of the orbit elements of the mission spacecraft subject to PLP thrust $F = 0.008$ with the initial excursion being $\mathbf{L}_0 = 0.1\,\mathbf{i}$. The units for $(i, \Omega, \omega)$ are degrees, whereas $a$ is dimensionless.

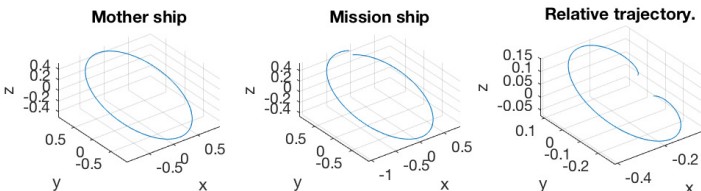

**Figure 10.** (**Left**) Trajectory of the mother ship; (**middle**) trajectory of the mission spacecraft; (**right**) relative trajectory of the mission spacecraft to the mother ship. The initial excursion is $\mathbf{L}_0 = 0.1\,\mathbf{j}$.

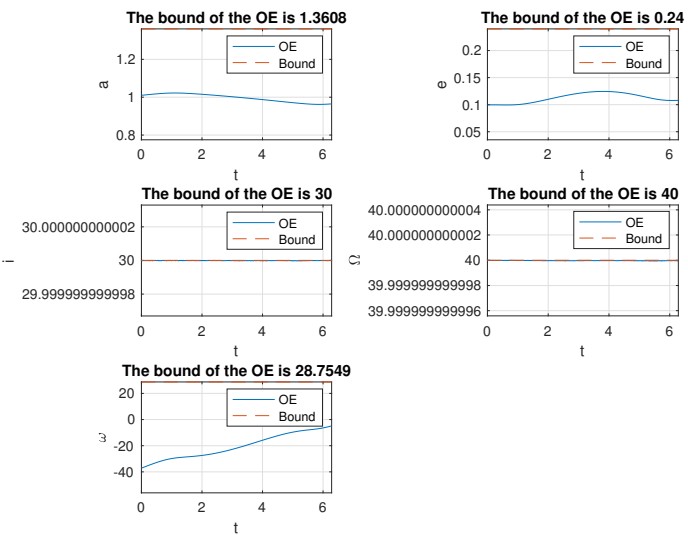

**Figure 11.** Variation of the orbit elements of the mission spacecraft subject to PLP thrust $F = 0.008$ with the initial excursion being $\mathbf{L}_0 = 0.1 \mathbf{j}$. The units for $(i, \Omega, \omega)$ are degrees, whereas $a$ is dimensionless.

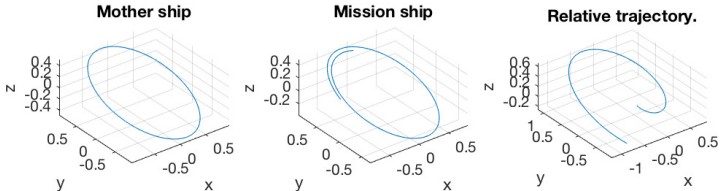

**Figure 12.** (**Left**) Trajectory of the mother ship; (**middle**) trajectory of the mission spacecraft; (**right**) relative trajectory of the mission spacecraft to the mother ship. The initial excursion is $\mathbf{L}_0 = -0.1 \mathbf{i}$.

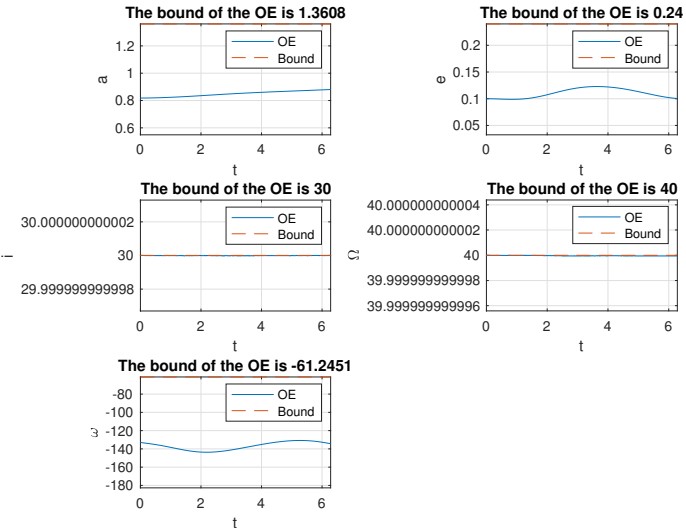

**Figure 13.** Variation of the orbit elements of the mission spacecraft subject to PLP thrust $F = 0.008$ with the initial excursion being $\mathbf{L}_0 = -0.1 \mathbf{i}$. The units for $(i, \Omega, \omega)$ are degrees, whereas $a$ is dimensionless.

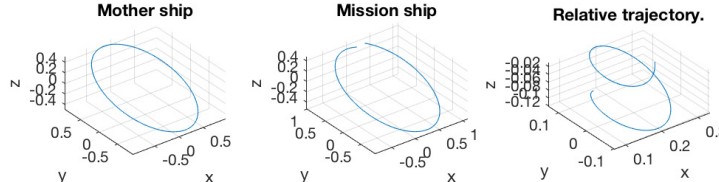

**Figure 14.** (**Left**) Trajectory of the mother ship; (**middle**) trajectory of the mission spacecraft; (**right**) relative trajectory of mission spacecraft to the mother ship. The initial excursion is $\mathbf{L}_0 = -0.1 \, \mathbf{j}$.

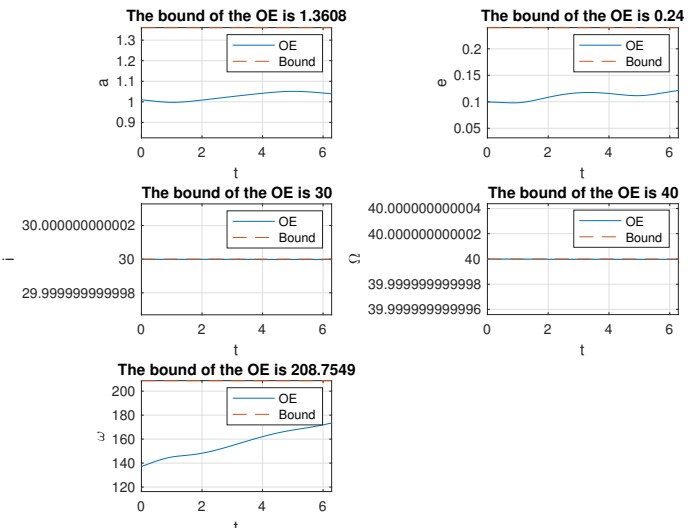

**Figure 15.** Variation of the orbit elements of the mission spacecraft subject to PLP thrust $F = 0.008$ with the initial excursion being $\mathbf{L}_0 = -0.1 \, \mathbf{j}$. The units for $(i, \Omega, \omega)$ are degrees, whereas $a$ is dimensionless.

### 7.2. Influence of a Small Planar PLP Thrust

This section verifies the sufficient condition of the lowest thrust required for interplanetary travel. According to the previous results, the mission spacecraft might be trapped in the vicinity of the mother ship if the normalized thrust force is less than 0.008.

Figures 16 and 17 verify the result. In these two simulations, the initial excursion $\mathbf{L}_0 = -0.1 \, \mathbf{j}$ was selected. The simulation was run for 200 normalized periods. It is obvious that the mission spacecraft stayed in the vicinity of the mother ship, either from the relative trajectory or the variation of the orbit elements. Notably, the bounds in the titles of Figure 17 are only valid for one normalized period.

Although the fourth case in the previous simulations was selected to demonstrate in this section, similar results can be obtained using different initial offsets.

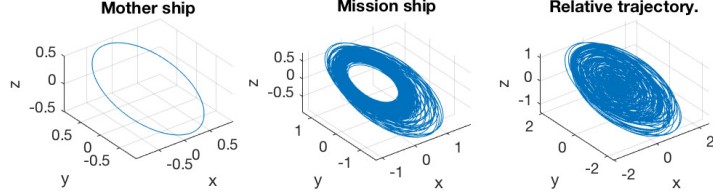

**Figure 16.** (**Left**) Trajectory of the mother ship; (**middle**) trajectory of the mission spacecraft; (**right**) relative trajectory of the mission spacecraft to the mother ship. The initial excursion is $\mathbf{L}_0 = -0.1 \, \mathbf{j}$. This simulation was run for 200 normalized periods.

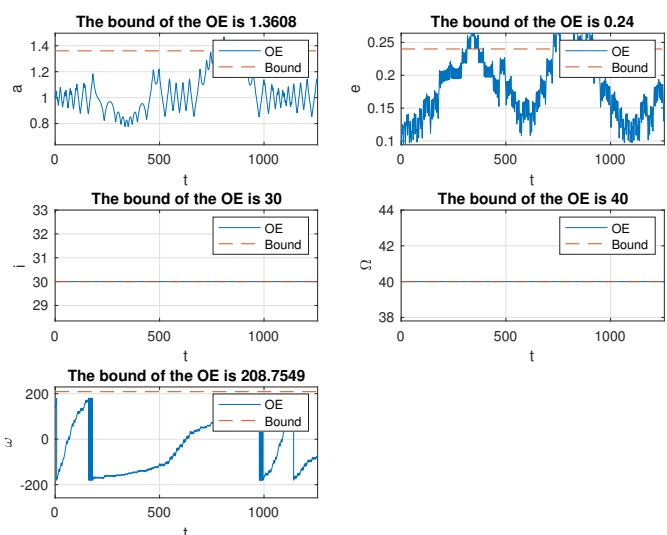

**Figure 17.** Variation of the orbit elements of the mission spacecraft subject to PLP thrust $F = 0.008$ with the initial excursion being $\mathbf{L}_0 = -0.1\,\mathbf{j}$. This simulation was run for 200 normalized period. The units for $(i, \Omega, \omega)$ are degrees, whereas $a$ is dimensionless. Notably, the bounds in the titles are only valid for 1 normalized period.

A counter example is presented in Figures 18 and 19. In this simulation, all parameters of the mission spacecraft were set identical to the previous simulation. There were only two differences. First, the thrust level was set as twice the previous, i.e., $F = 0.016$. Secondly, the simulation only ran for 25 normalized period because the trajectory diverged very fast after that.

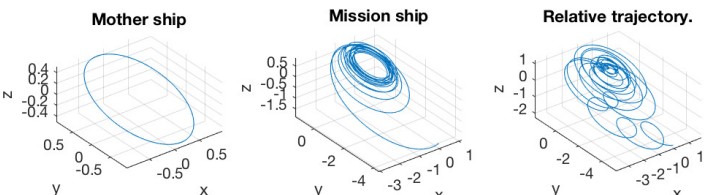

**Figure 18.** (**Left**) Trajectory of the mother ship; (**middle**) trajectory of the mission spacecraft; (**right**) relative trajectory of the mission spacecraft to the mother ship. The initial excursion is $\mathbf{L}_0 = -0.1\,\mathbf{j}$. This simulation was run for 25 normalized periods. Notably, the bounds in the titles are only valid for 1 normalized period.

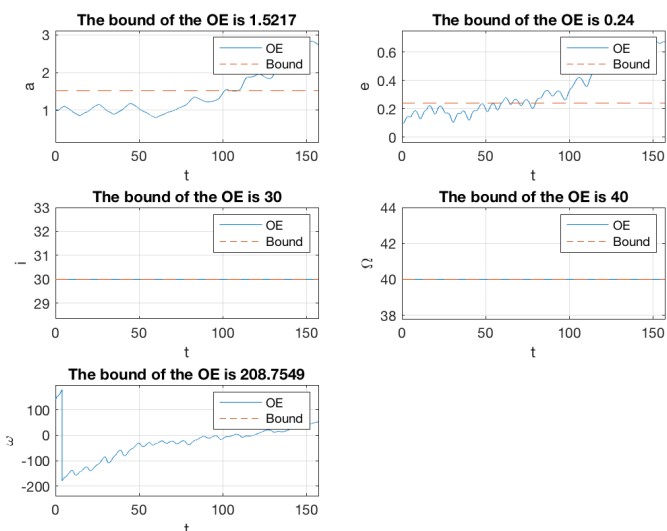

**Figure 19.** Variation of the orbit elements of the mission spacecraft subject to PLP thrust $F = 0.016$ with the initial excursion being $\mathbf{L}_0 = -0.1\,\mathbf{j}$. This simulation was run for 25 normalized period. The units for $(i, \Omega, \omega)$ are degrees, whereas $a$ is dimensionless.

### 7.3. Influence of a Small Out-of-Plane PLP Thrust

Figures 20 and 21 present the simulation results of an out-of-plane case. Most parameters were selected identical to the planar cases. The only difference was imposed on the initial offset, setting it as $\mathbf{L}_0 = 0.1\,\mathbf{k}$. Notably, the results derived in Section 6 cannot be employed to predict the simulation directly. Once the mission spacecraft is expelled by an out-of-plane thrust, the motion will generate a planar component naturally. This enforces the PLP force to have planar components, and the variation of orbit elements should consider both planar and out-of-plane perturbations. As a result, the simulation in this section only qualitatively demonstrates how a small out-of-plane PLP thrust influences the orbit elements of the mission spacecraft. In the simulation, all parameters were identical to the previous simulations.

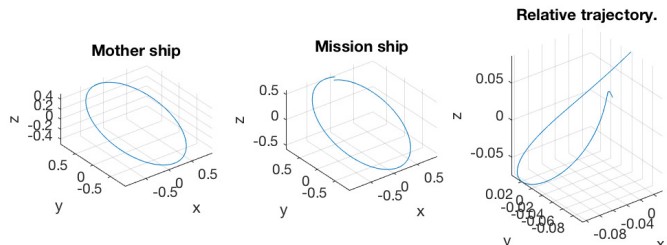

**Figure 20.** (**Left**) Trajectory of the mother ship; (**middle**) trajectory of the mission spacecraft; (**right**) relative trajectory of the mission spacecraft to the mother ship. The initial excursion is $\mathbf{L}_0 = 0.1\,\mathbf{k}$.

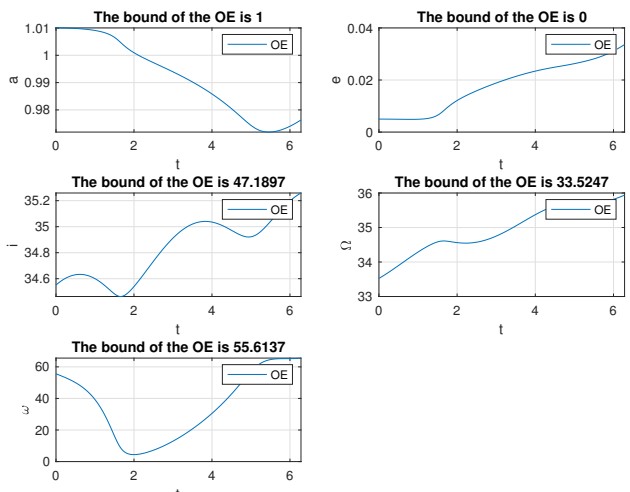

**Figure 21.** Variation of the orbit elements of the mission spacecraft subject to PLP thrust $F = 0.008$ with the initial excursion being $\mathbf{L}_0 = 0.1\ \mathbf{k}$.

### 7.4. Practical Applications

This simulation, presented in Figures 22 and 23, was to verify the proposed algorithms with full-dimensional parameters and study potential practical applications. In this simulation, a mother ship was assumed to be placed in a circular orbit of altitude 400 km above the Earth's surface. Other orbit elements were identical to the previous simulations. As for the PLP force, we had a laser of power 10 W, and the reflectance of the mirror installed on the mission spacecraft was assumed to be $R_m = 0.999998$. The mass of the mission spacecraft was assumed to be 1 kg. These three parameters were cited from our previous investigation in [9]. Suppose the mission spacecraft is lifted by a robotic arm for launch. The robotic arm was set as 17 m long, which was inspired by the Canadarm2 robotic arm installed on the International Space Station.

According to these settings, the PLP thrust can be computed as:

$$F_{PLP} = \frac{2PR_m S}{c} = 0.0334 \ (\text{N}) \tag{111}$$

The corresponded normalized force is given by:

$$\begin{aligned} F &= F_{PLP} \frac{R_m^2}{\mu_E} \\ &= 0.0334 \times \frac{6778137^2}{3.986 \times 10^{14}} = 0.0038 \end{aligned} \tag{112}$$

Moreover, the initial offset was assumed to be 17 m, corresponding to a normalized offset of:

$$l_0 = \frac{17}{6778137} = 2.5081 \times 10^{-9} \tag{113}$$

The simulation was run for 200 orbit periods, where one period was:

$$T = 2\pi \sqrt{\frac{6778137^3}{3.986 \times 10^{14}}} = 5553.6 \ (\text{s}) \tag{114}$$

It is obvious, from the simulation results, that the proposed algorithms provided a good prediction on the variation of the orbit elements of the mission spacecraft.

One thing to note is that the bound of $\omega$ was unreasonably high. If we look back at Equation (20), a singularity will exist in the variation equation of $\omega$ given $e = 0$, and large

error might exist for $e \approx 0$. In our simulation, $\bar{e} = 2.5081 \times 10^{-9} \approx 0$, and the singularity dominated the result. Hence, the bound of $\omega$ was actually meaningless in this case.

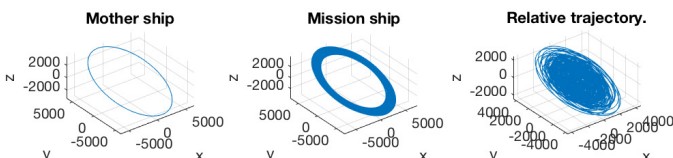

**Figure 22.** (**Left**) Trajectory of the mother ship; (**middle**) trajectory of the mission spacecraft; (**right**) relative trajectory of the mission spacecraft to the mother ship. The initial excursion is $\mathbf{L}_0 = 2.5081 \times 10^{-9}$ **i**. This simulation was run for 200 orbit period.

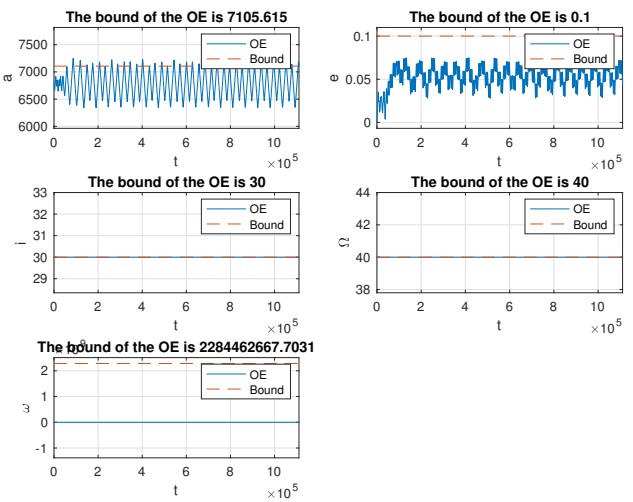

**Figure 23.** Variation of the orbit elements of the mission spacecraft subject to PLP thrust $F = 0.0038$ with the initial excursion being $\mathbf{L}_0 = 2.5081 \times 10^{-9}$ **i**. This simulation was run for 200 orbit period. The units for $(i, \Omega, \omega)$ are degrees, whereas $a$ is in kilometer. Notably, the bounds in the titles are only valid for 1 orbit period.

### 7.5. Formation Control by C-W Equations

An interesting example presented in this section is to control a spacecraft formation by the C-W equations using PLP thrust. The C-W equations are well known for the linearly approximated relative trajectory to a nominal circular orbit in the rotational frame. Let $\mathbf{L} = \zeta \hat{\mathbf{e}}_r + \eta \hat{\mathbf{e}}_\theta + \xi \hat{\mathbf{e}}_h$ and $\mathbf{L}' = \dot{\zeta} \hat{\mathbf{e}}_r + \dot{\eta} \hat{\mathbf{e}}_\theta + \dot{\xi} \hat{\mathbf{e}}_h$ be the relative velocity in the rotational frame. The linearized equations of motion with small PLP thrust by the C-W equations are:

$$\ddot{\zeta} = 3n_R^2 \zeta + 2n_R \dot{\eta} + F\frac{\zeta}{l}, \tag{115}$$

$$\ddot{\eta} = -2n_R \dot{\zeta} + F\frac{\eta}{l}, \tag{116}$$

$$\ddot{\xi} = -n_R^2 \xi + F\frac{\xi}{l}. \tag{117}$$

In the normalized system, the equations look similar by setting $n_R = 1$. The unperturbed C-W equations have closed-form solutions. One of the famous special cases is linear drift, which requires the initial conditions to be $\mathbf{L}_0 = (\zeta_0, \eta_0, 0)$ and $\mathbf{L}'_0 = (0, -1.5\zeta_0, 0)$.

In order to test the robustness of our result, the linear drift case was selected. In the simulation presented in Figure 24, the initial conditions were set as $\mathbf{L}_0 = (0.1, 0, 0)$ and $\mathbf{L}'_0 = (0, -0.15, 0)$. The simulation was run for five orbit periods. It is clear in the figure that an uncontrolled trajectory, presented by a dashed line, linearly drifts away. The controlled trajectory presented by a solid curve, however, is confined within a small region. One should realize that the controlled trajectory is not a closed curve, though it looks as an

ellipse. This example verifies the proposed PLP trust threshold and implies that this type of trust is potentially applicable to the formation flight of spacecraft.

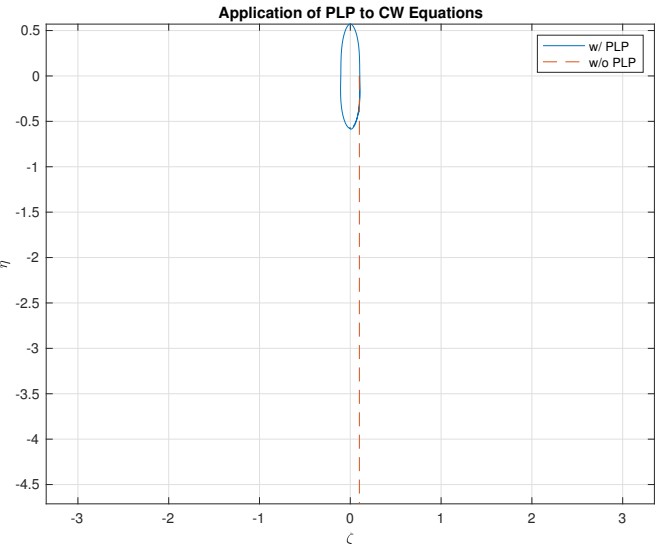

**Figure 24.** Variation of the orbit elements of the mission spacecraft subject to PLP thrust $F = 0.008$ with the initial excursion being $\mathbf{L}_0 = 0.1\,\mathbf{i}$. This simulation was run for 5 orbit period. $\zeta$ and $\eta$ are in normalized length.

## 8. Conclusions

This study investigated the variation of the orbit elements of a spacecraft propelled by photonic laser propulsion (PLP) under the two-body problem assumption. This study first reviewed some facts of PLP thrust and background knowledge of celestial mechanics. Perturbation theory was also reviewed and introduced to investigate this problem. Since PLP thrust is continuous and constant, the Gauss equations were introduced to study the variation of osculating orbit elements. In order to generalize the study, all equations were normalized based on the dynamics of the mother ship. The mission spacecraft subject to planar and out-of-plane PLP thrusts was studied, respectively. One should note that in practice, a pure out-of-plane PLP thrust for the whole mission is infeasible. The motion of the spacecraft will naturally cause planar components of the PLP force. As a result, a practical motion subject to initial out-of-plane PLP thrust will eventually be described by the combination of planar and out-of-plane effects. In this study, bounds on the variation of the orbit elements of the mission spacecraft were derived for both small planar and out-of-plane thrusts by zeroth- and first-order approximations. A sufficient condition that traps the mission spacecraft in the vicinity of the mother ship was also found. The study suggested that it may take a very long time period for the mission spacecraft to leave the mother ship if the normalized PLP thrust is less than 0.008. A similar, but more rigorous criterion was proposed in [9]. Our new discovery pushes the bound higher. This threshold was also verified by the linearized dynamics described by the Clohessy–Wiltshire equations. All proposed results, including the bounds and the sufficient conditions, were verified by numerical simulations. Our work presented in this paper is directly applicable to the usage of PLP thrust in future space missions, such as the formation flight about a circular orbit or interplanetary travel.

**Funding:** This research was partly funded by the Department of Education of Taiwan and partly by the National Science Council of Taiwan through Project NSC-100-2221-E-032-032.

**Conflicts of Interest:** The authors declare no conflict of interest. The funders had no role in the design of the study; in the collection, analyses, or interpretation of the data; in the writing of the manuscript; nor in the decision to publish the results.

## Appendix A. Approximation of 1/*r*

As shown in Figure 4, $r = \sqrt{1 + 2l_0 \cos\theta + l_0^2}$. Given $\theta = \theta_0$, the approximation of a function $f(r) = f(l_0)$ can be obtained through the Taylor expansion about the operation point $l_0 = 0$. Let $f(r) = 1/r = 1/\sqrt{1 + 2l_0 \cos\theta_0 + l_0^2} = f(l_0)$, and the function can be approximated by:

$$
\begin{aligned}
f(l_0) &\approx f(l_0)|_{l_0=0} + \left.\frac{\partial f}{\partial l_0}\right|_{l_0=0} l_0 + \frac{1}{2!}\left.\frac{\partial^2 f}{\partial l_0^2}\right|_{l_0=0} l_0^2 \\
&\approx 1 - l_0 \cos\theta_0 + (3\cos^2\theta_0 - 1)\frac{l_0^2}{2}
\end{aligned}
\tag{A1}
$$

Hence, the first-order approximation is found by $1/r \approx 1 - l_0 \cos\theta_0$.

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
