# Peer review of "Variation of Osculating Orbit Elements Using Low-Thrust Photonic Laser Propulsion in the Two-Body Problem"

_aerospace, doi:10.3390/aerospace9020075_

Round 1

Reviewer 1 Report

This paper analyzes the variations of orbital elements of a mission spacecraft propelled by the PLP system under a two-body assumption. An approximated upper bound thrust F that traps the mission spacecraft is provided. Numerical simulations are provided to validate the proposed result. However, there are several issues that need to be addressed in this paper before being accepted for publication.

Major Issues:

  1. pp. 11, the derivation of Eq. (64) is not mathematically solid. The conditions mentioned in line 234 cannot necessarily lead to the result of Eq. (64), especially under the integral. For example, -1<=cos(x)<=1. However, Integral[ cos(x) ]dx <= Integral[ 1 ]dx is not always true. Clear justification and derivations should be provided. The same issue also appears in the derivation of Eq. (74) and Eq. (83).
  2. pp. 14, Eq. (83): besides the issue mentioned in Comment 1, it is also not clear how the integral sign is removed for sqrt(K_w) when K_w is a function of E. This derivation is not solid at this moment.
  3. Clear cross-references for figures should be provided in the Numerical Simulation section. Otherwise, it makes the results hard to follow for readers. For example, subsection 7.4 doesn't provide any cross-reference link to Figs. 23 and 24, which are figures for the analysis in this section.

Minor Issues:

  1. pp. 4 lines 113-116: This paragraph needs to be rephrased to clearly express multiple assumptions in this paper. For example, "This study does not consider the reaction force by the PLP on the mother ship because in practical applications." This sentence is not complete.
  2. pp. 6 Eq (21), even if some parameters have been normalized, if mu appear on the left-hand side of the equation, it should not be omitted on the right-hand side. The same problem appears in Eq. (92)
  3. pp. 14, line 265, based on Fig. 5, F_f should be equal to F*sin(Phi) rather than sin(Phi).

Reviewer 2 Report

The PLP propulsion is a special and promising propulsion method.  This paper  investigates its  influences in two-body assumption.  Some conclusions are obtained in orbital mechanics. 

some suggestions are given 

(1)shorten this paper, for example the well known Figure 4 and Equation 6-11, etc. could be omitted.

(2)generate some special application example, for example show readers an interesting example in formation control by CW equations using PLP based on conclusions in this paper.

(3) some verbal changes needed, for example, in line 6 it should be "approaches"

Reviewer 3 Report

Please see the document attached.

Round 2

Reviewer 1 Report

The author has adequately addressed all revision comments. Recommend accepting in present form.